# Evaluation of Nitrate Soil Probes for a More Sustainable Agriculture

**DOI:** 10.3390/s22239288

**Published:** 2022-11-29

**Authors:** Amelia Bellosta-Diest, Miguel Á. Campo-Bescós, Jesús Zapatería-Miranda, Javier Casalí, Luis M. Arregui

**Affiliations:** 1Department of Engineering, Universidad Pública de Navarra (UPNA), Campus de Arrosadía, 31006 Pamplona, Spain; 2Institute for Innovation & Sustainable Development in Food Chain (IS-FOOD), Universidad Pública de Navarra (UPNA), Campus de Arrosadía, 31006 Pamplona, Spain; 3Department of Agricultural Engineering, Biotechnology and Food, Ed. Los Olivos, Universidad Pública de Navarra (UPNA), Campus de Arrosadía, 31006 Pamplona, Spain

**Keywords:** nitrate, soil sensor, electrical conductivity, nitrate solution, moisture content

## Abstract

Synthetic nitrogen (N) fertilizers and their increased production and utilization have played a great role in increasing crop yield and in meeting the food demands resulting from population growth. Nitrate (NO_3_^−^) is the common form of nitrogen absorbed by plants. It has high water solubility and low retention by soil particles, making it prone to leaching and mobilization by surface water, which can seriously contaminate biological environments and affect human health. Few methods exist to measure nitrate in the soil. The development of ion selective sensors provides knowledge about the dynamics of nitrate in the soil in real time, which can be very useful for nitrate management. The objective of this study is to analyze the performance of three commercial probes (Nutrisens, RIKA and JXCT) under the same conditions. The performance was analyzed with respect to electrical conductivity (EC) (0–50 mS/cm) and nitrate concentration in aqueous solution and in sand (0–180 ppm NO_3_^−^) at 35% volumetric soil moisture. Differences were shown among probes when studying their response to variations of the EC and, notably, only the Nutrisens probe provided coherent accurate measurements. In the evaluation of nitrate concentration in liquid solution, all probes proved to be highly sensitive. Finally, in the evaluation of all probes’ response to modifications in nitrate concentration in sand, the sensitivity decreased for all probes, with the Nutrisens probe the most sensitive and the other two probes almost insensitive.

## 1. Introduction

Agricultural systems are nitrogen (N) deficient throughout the world. Various plant molecules, such as amino acids, chlorophyll, nucleic acids, ATP, and phyto-hormones, that contain nitrogen as a structural part are necessary to complete biological processes including carbon and nitrogen metabolisms, photosynthesis, and protein production [1]. Nitrogen is involved in various critical processes such as growth, leaf area-expansion and biomass-yield production. Thus, the production and utilization of synthetic N fertilizers have played a great role in increasing crop yield and meeting the demand of population growth.

Nitrogen exists in the soil system in many forms and occasionally transforms very easily from one form to another. Most of the nitrogen available in the soil is unavailable for the plants and needs to be transformed into nitrate (NO_3_^−^) or ammonium (NH_4_^+^), the available forms for plant uptake. The routes that N follows in and out of the soil system are collectively called the nitrogen cycle. The N cycle is biologically influenced, and the biological processes are influenced by prevailing climatic conditions along with a particular soil’s physical and chemical properties. The main processes of the N cycle are fixation, mineralization, immobilization, nitrification, and denitrification. Depending on the dynamics of the cycle and the inputs (fertilizers) added, the amount of nitrate in the soil will vary. Nitrate is the most abundant source of N that is available for plants in cultivated soils, and it is the common form of nitrogen absorbed by plants growing in the field [2].

Nitrate pollution of water resources may originate from several different pathways, including point sources (such as wastewater effluents and intensive livestock farming) and diffuse sources (such as fertilizers, extensive livestock farming, and atmospheric deposition). Presently, nitrate pollution from agricultural diffuse sources is considered the main cause of groundwater degradation in the European Union (EU) [3,4,5] due to the high solubility of nitrates in water and low nitrate retention by soil particles. Thus, nitrate is prone to leaching to the subsoil layer and ultimately to the ground water if not taken up by plants or denitrified to N_2_O and N_2_. Nitrate can also be easily transported by surface runoff to streams. This can be detrimental to human health and pollute biological environments by accelerating eutrophication, which causes dramatic increases in aquatic plant growth and changes the ecosystem composition of plants and animals that live in the stream. Within the EU, the reduction and prevention of water pollution caused by nitrate from agricultural sources was addressed by the Nitrate Directive 91/676/EEC. This directive establishes that both surface freshwaters and groundwater should be considered affected by nitrate pollution when they contain more than 50 mg L^−1^ of nitrate. The recommended limit for nitrate in drinking water and to prevent the eutrophication of freshwaters is 25 mg L^−1^.

Correct fertilizer application rates of N are critical in preventing leaching and its subsequent adverse effects. The optimum dose of nitrogen that must be provided to a crop depends on the crop, the fertility of the soil and the intended outcomes for the crop, among other factors. Therefore, in most of the cases, the decision to use a certain dose cannot be made from the calculation of N utilization by the crop, as has been done for many years based on the classic recommendations for fertilization in agriculture [6]. When N inputs to the soil system exceed crop needs, there is a possibility that excessive amounts of nitrate may enter either ground or surface water. For this reason, proper fertilization is key and mismanagement can have unintended consequences. Therefore, monitoring and controlling the amount of nitrate ions in the soil is essential.

Procedures for nitrate analysis in liquid media are well-established, and several in situ and laboratory methods are available for nitrate assessment in water [7]. The assessment methodologies include electrochemical detection, chromatography, electromagnetic methods, biosensors, UV (ultraviolet) sensors, fiberoptic sensors [8], mid-IR spectroscopy [9] and laser-induced break down spectroscopy (LIBS) [10]. Although a wide range of options for analyzing nitrate concentrations in water are available, only a few methods exist to analyze the mineralizable N in the soil. The classical method consists of determining mineralizable N by soil sampling and subsequent laboratory analysis. However, these analyses are costly due to the time and resources needed to conduct a qualified soil sampling procedure (protocol) and complete laboratory sample analyses [11,12]. Rapid methods have been developed to estimate mineralizable N in the soil. The first one consists of soil sampling and a quick field analysis, not necessarily conducted in a laboratory, and is based on previously established soil-specific baseline parameters [13]. The second type consists of the installation of probes such as lysimeters that extract the solution from the soil matrix for subsequent rapid analysis by colorimetry or spectrometry. The most innovative probes are electrochemical sensors designed to be set directly into the soil (in situ) to record instantaneous measurements. Thus, several such sensors have become available.

Despite their promising performance, electrochemical sensors for soil quality assessment are not getting enough attention. This is due to lack of awareness among farmers and the lack of promotion efforts by agencies [14]. These probes can provide real-time, accurate information about the trend of nitrate concentrations at a moderate cost, which in turn will inform optimal fertilization, thus reducing the environmental impact caused by nutrients into surface and groundwaters.

Therefore, the main objective of this study is to evaluate the performance of three different probes purchased in 2019: Nutrisens sensor (Verde Smart Co., Huelva, Spain) [15], NPK sensor RK520-05 (Hunan Rika Electronic Tech Co., Changsha, China) [16] and NPK sensor (JXCT, Weihai, China) [17]. The probes were first tested in solutions with a range of electrical conductivities (EC) from 0 to 56 mS/cm in the absence of nitrate ions, then in an aqueous solution of nitrate ion between 0 and 180 mg/L, and finally in almost saturated sand (at 35% volumetric moisture) with different nitrate ion solutions from 0 to 150 mg/L, as the first proxy of the soil. The range of concentrations chosen is within the usual range for crop fields, considering that the concentration of nitrate in the soil can be highly variable depending on management practices [18,19,20]. For statistical analysis, repeatability and reproducibility indexes have been used in the study to represent the variability of the readings [21]. These indexes were conducted by a one factor Analysis of Variance (ANOVA) test and are the usual ones for the manufacturing of sensors [22]. To date, to the best of our knowledge, the few nitrate sensors available have not been evaluated under the same experimental setup.

## 2. Materials and Methods

### 2.1. Sensors

The main characteristics of the sensors evaluated are described below, in Figure 1.

The Nutrisens sensor [15] measures nitrate and potassium ions and is based on electrochemical detection. Electrochemical sensors are based on several electrochemical reactions at the surface of an electrode. This probe transforms the electrochemical signal into a voltametric signal [23,24,25]. The Nutrisens sensor measures the nutrient concentration trend and not the nutrient concentration. Due to the heterogeneity of the soil, it cannot be calibrated reliably, but it can be calibrated in solution. Its development and patent [26] by the Spanish company Verde Smart with the Autonomous University of Barcelona (UAB) took several years to achieve.

The Nutrisens sensor is based on an Ion Selective Electrode (ISE). According to the recommended definition of the International Union of Pure and Applied Chemistry (IUPAC), an ISE is an electrochemical sensor whose potential response has a linear relationship with the logarithm of the specific ionic activity in the solution. An ion selective membrane is the key component of an ion selective electrode. Due to the differences of ionic activity (concentration), ion exchange happens on the surface of both sides of an ion selective membrane to form membrane potential [27].

These sensors require a linear and a nonlinear adjustment during preparation and evaluation. Both adjustments are logarithmic, but the non-linear adjustment uses all the points (Nikolskii-Eisenman equation), whereas the linear adjustment (Nernst equation) only considers the linear part of the logarithmic fitting and is the one used in the evaluation of the probes [27].

The Nutrisens sensor is made up of the probe and signal adaptation electronics as can be seen in Figure 2.

The lower part of the black face in the probe is the sensor area, which contains a gelatinous material that is the most sensitive part, and although it is protected, precautions must be taken not to hit this area during installation. The probe must be connected to the signal adaptation electronics through the corresponding connector. It can be connected in analog or digital mode. For the digital mode, which was used during this experiment, the SDI12 module, also known as SDIAN, must be connected between the signal adaptation electronics of each of the probes and the datalogger Campbell CR10X.

The Rika (Hunan Rika Electronic Tech Co., Changsha, China) [16] and JXCT (JXCT, Shandong, China) [17] are suitable for detecting the content of nitrogen, phosphorus, and potassium. No documentation was available to determine the physical principle of measurement. Information was requested from the manufacturers but has not been provided. Based on Longhurst & Nicholson [28], we assume that these probes are conductivity meters that measure the EC to derive the concentration of the ions using an equation. Both sensors have similar measurement parameters, the communication protocol is RS485 carried out with Arduino, and they are powered with a 12V DC power supply. The probes measurement units are mg/kg and they are default calibrated by the company for a nitrate concentration between 500 and 1900 mg/kg (personal communication from JXCT). The calibration equation is unknown and embedded by the manufacturer in the probe, so the raw electrical measurement signal is not available. The detection range is between 0 and 1999 mg/kg, as can be seen in the manufacturer’s manual [29].

### 2.2. Evaluation of the Probes’ Signal Response to the Variation of Electrical Conductivity (EC) in Liquid Medium in the Absence of Nitrate Ions

This test consists of evaluating the effect of the probe ion signal (output signal) by varying the Electric Conductivity (EC) in the absence of nitrate. EC is modified by adding NaCl (Table 1). One liter of deionized water (from which the cations and anions have been extracted to prevent them from influencing the sensor readings) was used and increments of 5 g of sodium chloride (NaCl) up to 35 g corresponding to the salinity of seawater were added. Table 1 shows the electric conductivity of each solution measured with a conductivity meter (CRISON GLP32).

Measurements were then taken, as seen in Figure 3. One Nutrisens probe (only one was accessible at the time) and three each of the RIKA and JXCT probes were used. One reading was made at each EC solution. The probes need to stabilize in the medium, so for the JXCT and RIKA the measurements were taken at one minute intervals, and for the Nutrisens probe 5 min intervals were selected according to the manufacturer’s recommendations for the probe membrane to reach equilibrium. The probes were cleaned with deionized water in between measurements of different concentrations.

### 2.3. Evaluation of the Probes’ Signal Response to the Variation of Nitrate Concentration in Liquid Medium

The objective of this test is to evaluate the sensors under controlled laboratory conditions in a liquid medium using deionized water solutions and a range of standard nitrate solutions. As a starting material, three solutions of nitrogen nitrate were used (44 mg NO_3_^−^/L, 100 mg NO_3_^−^/L y 1000 mg NO_3_^−^/L) (HACH^®^ and Supelco^®^), which were diluted into 12 different concentrations between 0 and 180 mg NO_3_^−^, corresponding to the range of agronomic interest.

Measurements were then taken (see Figure 4). One probe of each type was used, and five readings were made at every nitrate concentration solution. 

### 2.4. Evaluation of the Effect of Nitrate Concentration on the Probes’ Signal Response in Almost Saturated Sand, a Soil Proxy

The objective of this work is to evaluate, for the first time under controlled conditions, the effect of the nitrate ion concentration on the dissolution of the soil matrix on the electrical signals of the probes analyzed. To simplify and allow a clear interpretation of these first results, sand was used as a proxy for soil. 

The type of sand used during the test was a siliceous sand with a grain size ranging from 0.2 to 0.7 mm. This type of material was chosen due to its homogeneity, lack of structure, and ease of handling so that the added solution could be distributed homogeneously.

First, sand samples were prepared. The processes started with drying the sand in the oven at 110 °C for 24 h to obtain a moisture content of 0%. Once the sand was dried, the volumetric humidity of the sand for the different moisture conditions was determined as the measurement reference. This is calculated from the ratio between the volume of the liquid fraction and the volume of the sample, applying the following formula:θ = Vw/Vs,(1)
where: θ is the volumetric soil moisture (%); Vw is the water volume (mL) and Vs is soil sample volume (mL).

To calculate the humidity, one liter of water was introduced into a container and that water level was marked with a sharpie. The water was then removed, the container was dried and weighed, and the sand was poured up to the mark to represent equal volume. The weight of the sand, in this case 1400.5 g, was recorded to add the same amount to all containers, as seen in Figure 5.

According to [30], the measured capacitance is altered by the soil temperature and the temperature of the critical components in the measurement circuits. This factor must be considered since the sand was at 110 °C in the oven, so it must be cooled for 24 h so that it reaches room temperature.

The second step was to prepare the dissolutions. Three nitrogen nitrate solutions were used as a starting material (44 mg NO_3_^−^/L, 100 mg NO_3_^−^ N/L y 1000 mg NO_3_^−^/L) (HACH^®^ and Supelco^®^), and they were diluted at different concentrations between 0 and 150 mg NO_3_^−^·L^−1^ (agronomic interest range) using deionized water.

The studied concentrations were 10, 25, 50, 75 and 150 ppm. One liter of each solution at each concentration was prepared to increase the humidity of the sand. To confirm that the solutions were made correctly, a nitrate meter from HORIBA^®^ [31] was used. The meter was successfully calibrated in advance with two standard solutions, 150 ppm and 2000 ppm, and proven to be accurate post-calibration.

Once the containers with the dried soil and the solutions were prepared, the volumetric humidity in the different sand samples was measured. The evaluation of the probes was made at 35% volumetric humidity (sand saturation) for an initial approach because there is more solution in the matrix, and this provides an optimal environment for probe performance. The sand was compacted for every trial in order to maintain the same volume.

Three probes of each type were used and three measurement readings per probe and trial were done, as seen in Figure 6. The Nutrisens probe readings were obtained by inserting the probe, compacting the soil around the probe, and then waiting 5 min until probe stabilization occurred. For the JXCT and Rika probe readings, the sand was first compacted and then the probe was inserted to the point where the sensor part was completely covered. Readings were taken after 15 s.

### 2.5. Calibration

To accurately measure nitrate concentrations, these sensors need a soil-specific calibration due to differences in clay content and cation exchange capacity of each soil type. Therefore, standard formulas to estimate the nitrate concentration are not reliable, and it is advisable to calibrate the probe for each soil type. The calibration method will be linear or non-linear depending on the response dynamics of the probes and the adjustment method will be by least squares.

The Nutrisens sensor provides readings in millivolts and requires a calibration equation for each single probe, according to the manufacturing company, because the sensor membrane varies from one probe to the other. 

The RIKA and JXCT commercial sensors are provided with a general calibration equation which converts the electrical response between the electrodes into mg per liter. However, to enhance accuracy it is usually recommended to perform a soil-moisture specific calibration, since this calibration equation has been built based on a standard nitrate liquid solution.

During calibration, three probes of each type were used to determine the repeatability and reproducibility of results. Three repeated measurements were performed in each trial. For the repetition, the probes were not extracted from the soil to avoid changing the compaction of the soil. The Gauge Repeatability and Reproducibility (R&R) study (see Figure 7) was conducted by a one factor Analysis of Variance (ANOVA) test, performed to analyze the variability of data between probes and within a single probe [32]. 

Repeatability is the variation of the results of several measurements obtained with successive attempts, with the same sensor and the same laboratory conditions. Reproducibility is the variation in the average of measurements made by different sensors using the same measurement system, measuring the same characteristics under the same conditions. Moreover, variability estimators associated with repeatability and reproducibility have been defined:(2)$$\sigma_{rpt}=\sqrt{MS_{ii\;}}$$
(3)$$\sigma_{rpr}=\sqrt{\frac{\left|MS_{ij\;}-MS_{ii\;}\right|}{n}}$$
(4)$$\sigma_{sensor}=\sqrt{\sigma^{2}{}_{rpt}+\sigma^{2}{}_{rpr}}$$
(5)$$\%\;Rpt=100\frac{\sigma^{2}{}_{rpt}}{\sigma^{2}{}_{sensor}}$$
(6)$$\%\;Rpr=100\frac{\sigma^{2}{}_{rpr}}{\sigma^{2}{}_{sensor}}\;\;\;\;$$
where: *σ_rpt_* and *σ_rpr_* are the variability estimators corresponding to repeatability and reproducibility respectively; *σ_sensor_* is the general variability associated to the sensor; *MS_ii_* is the mean square error of the measurements of the same sensor; *MS_ij_* is the mean square error among the measurements of the sensors; n is the number of observations and *% Rpt* and *% Rpr* are the corresponding percentages of the whole variation associated with repeatability and reproducibility, respectively [21,33].

## 3. Results and Discussion

### 3.1. Evaluation of the Probes’ Signal Response to the Variation of Electrical Conductivity (EC) in Liquid Medium in the Absence of Nitrate Ions

To assess the response of the probes in the absence of ion nitrate in a liquid medium, Figure 8 shows the response of the nitrate reading of the probes with respect to the increase in EC. 

For the Nutrisens probe, the results range from 225 to 360 mV and have a logarithmic trend, indicating that when the EC is low the probe has more sensitivity than when the EC is higher. Irrigation water has low EC, typically between 0 and 3 mS/cm [34]. However, soil salinity must be considered, since the probes are designed to work in the soil solution, and this can range from 0 to more than 16 mS/cm for saline soils [35]. It should be mentioned that these types of sensors are ion selective but not specific. That is, they are selective, in this case, to nitrate, but if there are ions with similar chemical characteristics, for example chloride, there is some interference and results may be affected. Chloride is the main interferent of this sensor. This interference is more pronounced if the ion to which it is selective is not present in the medium. According to the manufacturer, if the sensor is measuring nitrate, the results should trend on a decreasing logarithmic scale. In this case, it is the opposite, thus confirming that they are not measuring nitrate concentration when the EC is increasing, and that these membranes are not selective to nitrate ions.

For the Rika and JXCT probes, three probes each were analyzed. The results do not follow any pattern. It can be concluded that the probes are highly sensitive to EC, are not accurate, and for an EC higher than 40 mS/cm the probes stabilize, predicting a value of 208 NO_3_^−^ mg per liter. For the salinity range in agronomy, the Rika and JXCT probes are sensitive to EC and the results are not consistent (with an increasing or decreasing monotonic trend) between probes or companies.

Table 2 shows the fitting and determination coefficients for the different probes. The Nutrisens probe presents a perfect fit, while the results of the JXCT and Rika probes show their lack of consistency (or coherence), with no clear trend and random results up to 40 mS/cm, after which no sensitivity is shown.

### 3.2. Evaluation of the Probes’ Signal Response to the Variation of Nitrate Concentration in Liquid Medium

Table 3 shows the fitting and determination coefficients of the different probes when increasing the nitrate concentration in the liquid medium. Figure 9 represents the observed NO_3_^−^ mg/L versus the estimated measurements corresponding to each probe. As can be seen for the Nutrisens probe, the measured data show a decreasing logarithmic adjustment when increasing the nitrate concentration, which is consistent with what is presented in the patent [16]. The linear coefficient of determination (R^2^) is higher than 0.99. For the Rika and JXCT probes, the linear fit of both probes is close to the theoretical fit (1:1) and their linear coefficient of determination is similar and high around 0.96. All coefficients of determination can be said to be very acceptable, especially the one of the Nutrisens probe, which is practically 1. 

### 3.3. Evaluation of the Effect of Nitrate Concentration at 35% Soil Moisture on the Electrical Signals of the Probes Analyzed (Installed in Sand, a Soil Proxy)

Table 4 shows the fitting and determination coefficient for the Nutrisens probes when increasing the nitrate concentration in sand at 35% volumetric soil moisture. Figure 10 presents the electrical reading by the Nutrisens probes (millivolts) versus the observed nitrate concentration in the soil matrix solution. As can be seen for the Nutrisens probe, the reading output data have a decreasing power fitting when increasing the nitrate concentration, which is consistent with what is presented in the patent [27]. Probes 2 and 3 seem to have a better adjustment than Probe 1, which may be a consequence of the membrane deterioration. Throughout the experiment, it was observed that the gel covering the sensor was gradually damaged due to the high number of repetitions carried out during the test. According to the manufacturer, the probe is intended to be installed in one place during the sampling campaign, meaning that they are not designed to be reinstalled as the membrane can be damaged. This may have influenced the data collection.

To study the variability among readings of the same sensor and between different sensors of the same type, the Gage Repeatability and Reproducibility study using the ANOVA test was made for the Nutrisens probe. The results are shown in Table 5. The results indicate that all the readings are consistent for the different nitrate concentrations. In fact, all the *p* values are smaller than 0.05 which means that there is variability among the three probes for all the readings and the standard deviation of the readings among a single probe (repeatability) is very low, since the maximum deviation is 4. The standard deviation among the different probes (reproducibility) is very high, around 90, which causes the standard deviation of the sensor to be very high.

The variability among repeatability and reproducibility is distributed as a percentage. The results produce almost 0% repeatability, which indicates that there is hardly any variability between readings from the same probe, which is a positive aspect. On the other hand, reproducibility results are 100%, which indicates that there is high variability between probes. These results make sense since Nutrisens indicates that each probe must be analyzed individually, as they will not provide the same output (millivolts). 

The results of the RIKA and JXCT probes are represented in Figure 11 and Figure 12, respectively. As both companies have the same type of sensor, the results will be discussed together. The observed nitrate concentration is displayed against the estimated nitrate concentration provided by each probe. Observing the two figures, all of the results have the same trend, i.e., an increasing monotonic trend. This can also be seen in Table 6 and Table 7, where the coefficient “a” in the fitting indicates the slope of the linear fit. The smaller the value of the slope, the less variability between the results, and the values are far from the theoretical 1:1 perfect fit.

These probes are already calibrated by the manufacturers, but the equations implemented by the manufacturer are far from an adequate fit for the test performed. With a recalibration equation the results would most likely improve, although these probes are not very sensitive to nitrate variability since the slope of the linear regression is very small.

To study the variability among readings of the same sensor and among sensors, we proceeded to perform the Gage Repeatability and Reproducibility study using the ANOVA test in Excel. However, the ANOVA test was impossible to perform for these probes because there is no variability among readings of the same probe. This means that all the variability is caused by using different probes. Observing Figure 11 and Figure 12, the measured values are around the same interval, but there are some variabilities. For these probes it is also recommended to recalibrate them individually, even though the manufacturer does not indicate anything about recalibrating the probes.

## 4. Conclusions

In this work, the behavior of several probes developed and commercially manufactured by three companies for estimating nitrate concentration in the soil matrix have been studied under the same conditions.

First, the response of the probes to electrical conductivity (EC) in liquid medium in the absence of a nitrate concentration was studied. The Nutrisens probe shows an inverse relationship between the reading (millivolts) and the EC. This sensor is ion selective but not 100% specific to nitrate, and in this work, chlorine is the main interference ion, with interference exacerbated in the absence of nitrate. It has a variability of 130 mV from 0 to 55 mS/cm. The RIKA and JXCT probes do not follow any pattern when the EC is low. Below 40 mS/cm, each probe gives a different reading from the other probes, indicating they lack reliability and accuracy, and the measurements have no clear pattern.

The behavior of the probes was then studied in a liquid medium at different nitrate concentrations. In this test, the Nutrisens probe fits a power regression with an R^2^ greater than 0.99. The RIKA and JXCT probes produce a linear behavior, with linear regressions with R^2^ values of 0.95 and 0.96, respectively. It can be concluded that all three probes perform well in liquid media, with an emphasis on the Nutrisens probe’s slightly better performance. It is notable that the RIKA and JXCT probes have been calibrated by the manufacturer in solution and retain acceptable performance.

Finally, the behavior of the probes was studied in a sand matrix at 35% volumetric soil moisture and different nitrate concentrations. In this test, the Nutrisens probes have an electrical reading (millivolt) with a negative power trend as the nitrate concentration increases, and the trend is clear. According to the manufacturer, the probe is for determining the trend of the nitrate concentration, not to measure the exact nitrate concentration. This could be achieved by calibrating each probe individually. The RIKA and JXCT probes’ sensitivity is very low since the slope of the linear regressions is far from 1:1. These probes are already calibrated by the manufacturer; however, they could be recalibrated to achieve acceptable NO_3_^−^ estimation although the sensitivity would be very low.

This study provides the first assessment of these probes against one another in reproducible laboratory conditions. However, the probe response in solution with other macronutrients, such as potassium or phosphorus, should be analyzed, as well as a further study on the effect of EC and different volumetric soil moisture contents which were not considered in this study. In addition, it would be useful if this type of study is extended to other types of sand and soils with varying compositions to analyze the cross-matrix effects.

## Figures and Tables

**Figure 1 sensors-22-09288-f001:**
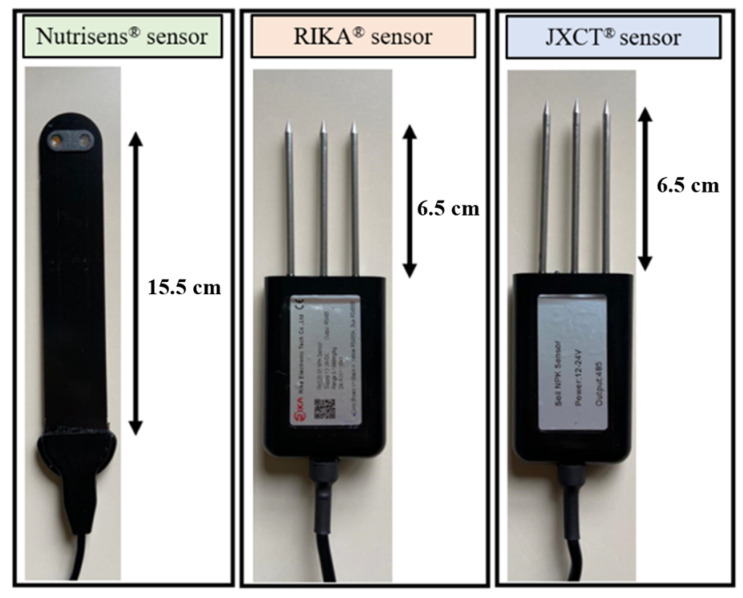
Picture of the sensors used in the study: Nutrisens, Rika and JXCT.

**Figure 2 sensors-22-09288-f002:**
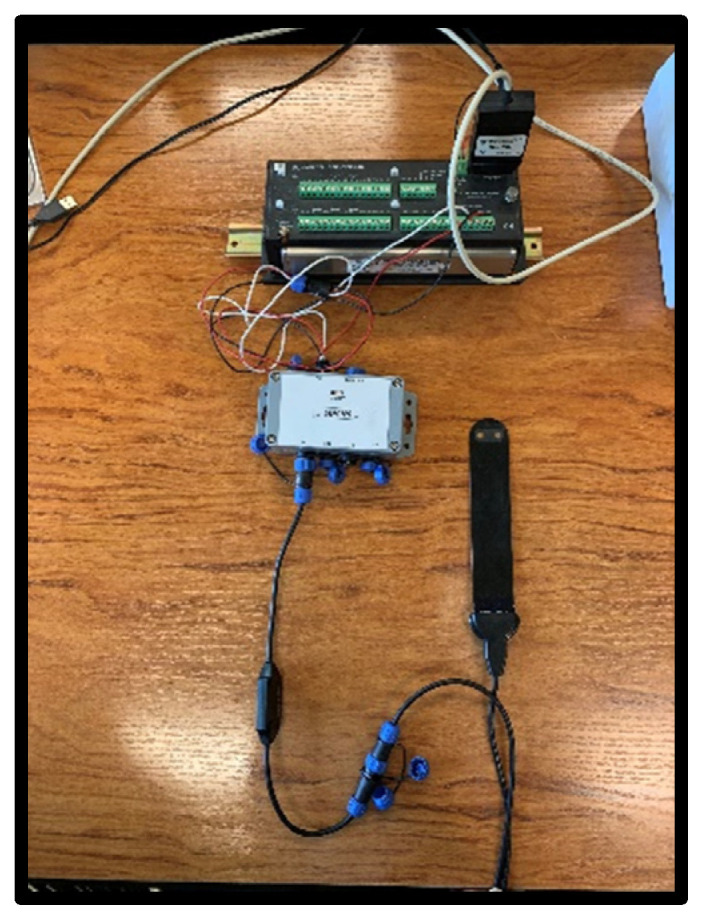
Nutrisens sensor connection with signal adaptation electronics, SDIAN and Campbell CR10X.

**Figure 3 sensors-22-09288-f003:**
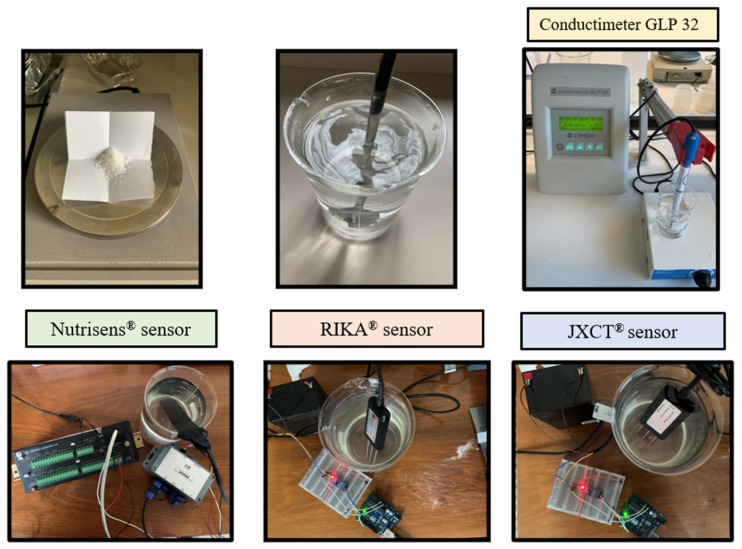
Experimental set-up for analyzing the effect of EC in the probes’ signal response in the absence of nitrate.

**Figure 4 sensors-22-09288-f004:**
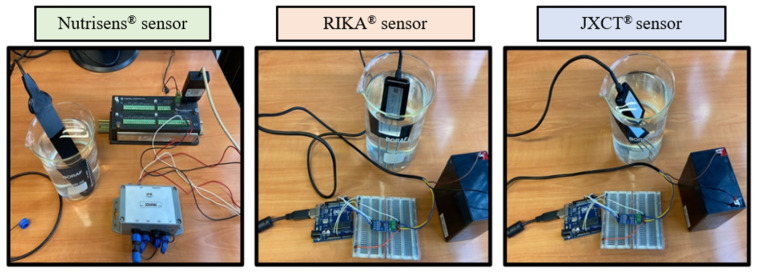
Experimental set-up of the three probes in nitrate dissolutions.

**Figure 5 sensors-22-09288-f005:**
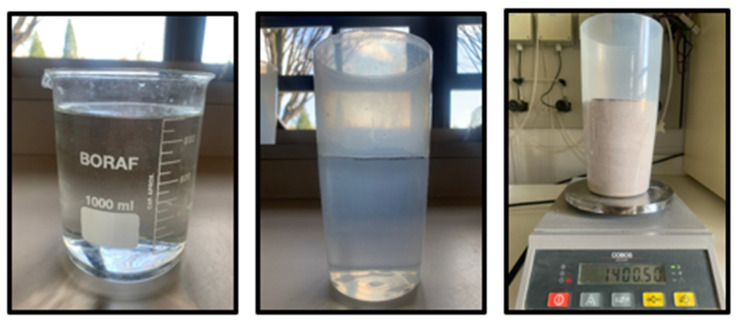
Steps for determining the volumetric moisture. One liter of water was measured and poured into the container where the probe measurements were placed. That volume was marked with a sharpie and filled with sand.

**Figure 6 sensors-22-09288-f006:**
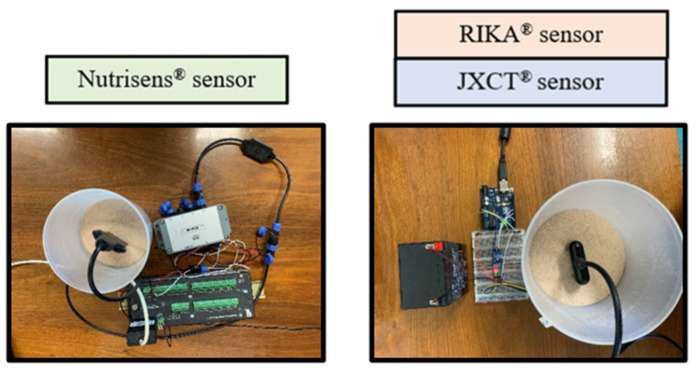
Process of reading of the probes in sand.

**Figure 7 sensors-22-09288-f007:**
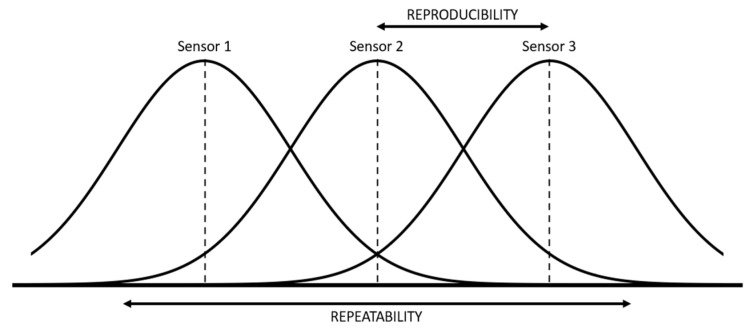
Graphical representation of the concept of repeatability and reproducibility.

**Figure 8 sensors-22-09288-f008:**
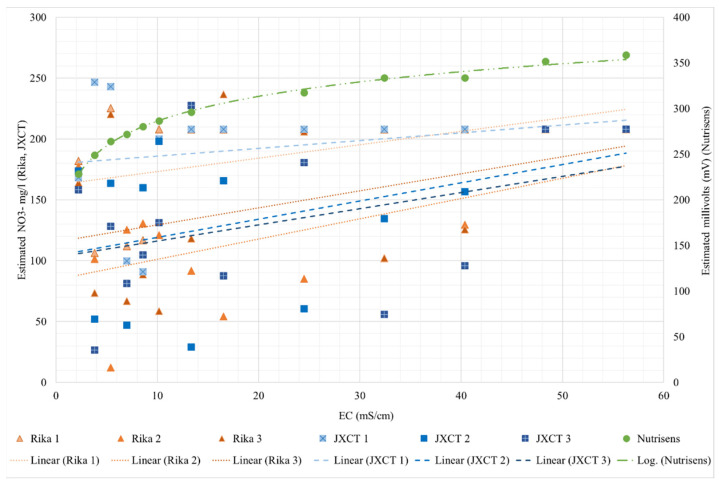
Behavior of the different probes (Nutrisens probe, RIKA and JXCT) when increasing the EC. On the *x*-axis the measured EC is shown; on the *y*-axis on the left the estimated NO_3_^−^ mg/L for the Rika and JXCT probes is shown, whereas on the right the estimated millivolts for the Nutrisens probe can be seen.

**Figure 9 sensors-22-09288-f009:**
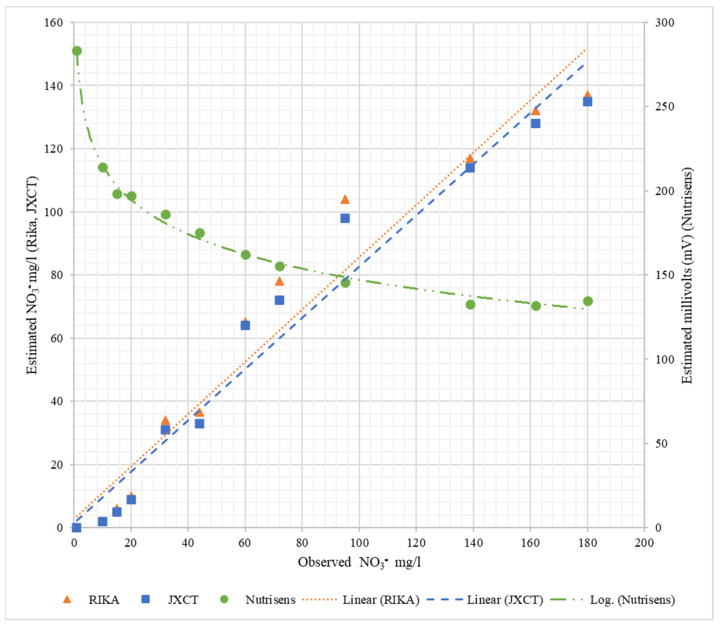
Behavior of the probes at different nitrate concentrations in liquid medium. On the *x*-axis is the observed nitrate concentration, on the left *y*-axis is the estimated nitrate concentration for Rika and JXCT, and on the right *y*-axis is the estimated millivolts for Nutrisens.

**Figure 10 sensors-22-09288-f010:**
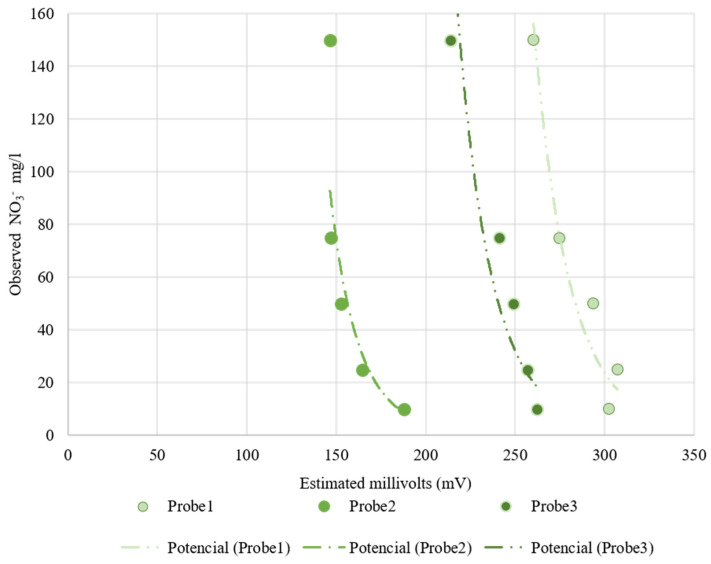
Nutrisens probes’ behavior at increasing nitrate concentration at a 35% volumetric soil moisture. On the *x*-axis is the estimated millivolts reading by the probes and on the *y*-axis is the observed nitrate concentration.

**Figure 11 sensors-22-09288-f011:**
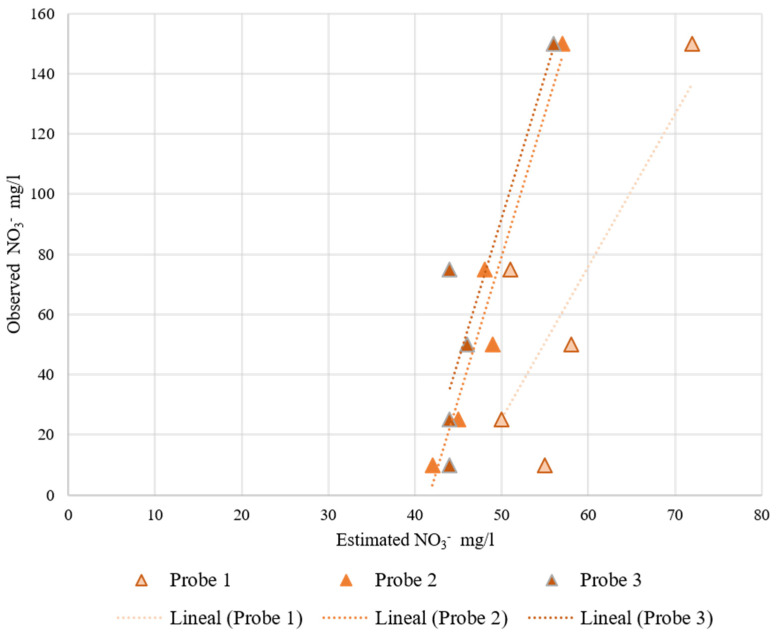
RIKA probes’ behavior at increasing nitrate concentration at a 35% volumetric soil moisture. On the *x*-axis is the estimated nitrate concentration measured by the probes and on the *y*-axis is the observed nitrate concentration.

**Figure 12 sensors-22-09288-f012:**
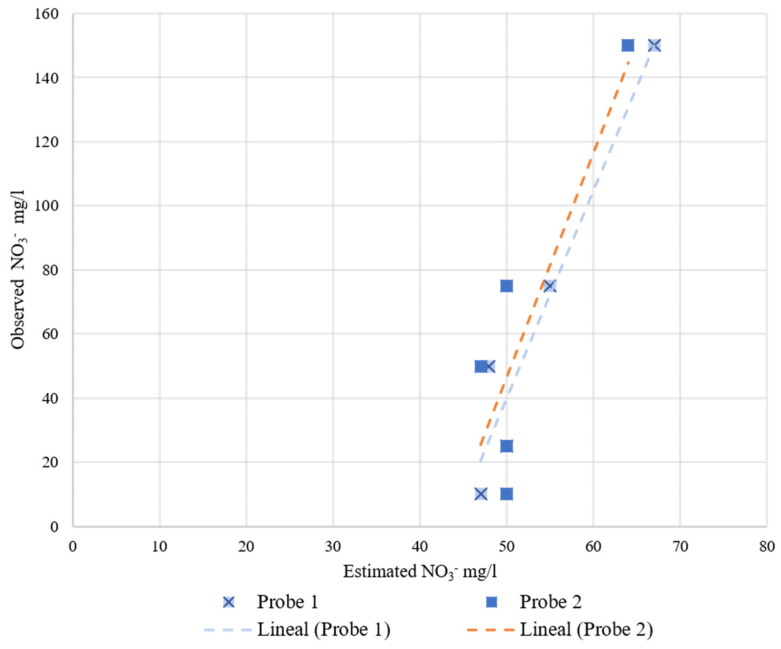
JXCT probes’ behavior at increasing nitrate concentration at a 35% volumetric soil moisture. On the *x*-axis is the estimated nitrate concentration measured by the probes and on the *y*-axis is the observed nitrate concentration.

**Table 1 sensors-22-09288-t001:** Amount of NaCl added to one liter of deionized water and the corresponding electric conductivity (EC).

Ec (mS/cm) ^1^	Amount of NaCl (g) ^2^
2.22	1
3.81	2
5.40	3
6.99	4
8.58	5
10.17	6
13.35	7
16.53	10
24.47	15
34.42	20
40.36	25
48.31	30
56.26	35

^1^ ±0.0001; ^2^ ±0.01.

**Table 2 sensors-22-09288-t002:** Fitting and determination coefficients for the probes when increasing the EC in absence of nitrate concentration.

Probe	Equation *	a	b	R^2^
Nutrisens	1	39.03	196.40	0.99
Rika	2	1.10	162.33	0.22
	2	1.67	84.53	0.29
	2	1.40	115.53	0.15
JXCT	2	0.64	179.40	0.06
	2	1.50	104.21	0.18
	2	1.33	102.83	0.15

* 1 represents y = aln(x) + b (where y is the estimated millivolts, x is the electric conductivity, a and b are the regression coefficients) and 2 is y = ax + b (where y is the estimated NO_3_^−^ mg/L, x is the electric conductivity, a and b are the regression coefficients).

**Table 3 sensors-22-09288-t003:** Fitting and the determination coefficients for the probes when increasing the nitrate concentration in liquid medium.

Sensor	Equation *	a	b	R^2^
Nutrisens	1	−29.56	283.26	0.99
RIKA	2	0.83	2.89	0.95
JXCT	2	0.81	1.42	0.96

* 1 represents y = aln(x) + b (where y is the estimated millivolts, x is the observed NO_3_^−^ mg/L, a and b are the regression coefficients) and 2 is y = ax + b (where y is the estimated NO_3_^−^ mg/L, x is the NO_3_^−^ mg/L, a and b are the regression coefficients).

**Table 4 sensors-22-09288-t004:** Fitting and determination coefficient for the Nutrisens probes when increasing the nitrate concentration in sand at 35% volumetric soil moisture.

Nutrisens Probe	a	b	R^2^
	y = ax^b^		
Probe 1	4 × 10^22^	−9.53	0.79
Probe 2	3 × 10^29^	−11.68	0.91
Probe 3	2 × 10^34^	−13.25	0.96

**Table 5 sensors-22-09288-t005:** Gauge Repeatability and Reproducibility study for Nutrisens probes.

NO_3_^−^	*p* Value	*σ_Repeatability_*	*σ_Reproducibility_*	*σ_Sensor_*	*%Repeatability*	*%Reproducibility*
10	5.77 × 10^−10^	4.11	82.16	82.27	0.25	99.75
25	8.92 × 10^−13^	1.73	102.13	102.14	0.03	99.97
50	1.71 × 10^−16^	0.41	101.97	101.97	0.00	100.00
75	3.50 × 10^−18^	0.20	93.71	93.71	0.00	100.00
150	4.40 × 10^−14^	0.83	80.89	80.90	0.01	99.99

**Table 6 sensors-22-09288-t006:** Fitting and determination coefficient for the RIKA probes when increasing the nitrate concentration in sand at 35% volumetric soil moisture.

RIKA Probe	a	b	R^2^
	Y = ax + b		
Probe 1	5.06	−227.45	0.67
Probe 2	9.49	−395.29	0.94
Probe 3	9.49	−381.91	0.81

**Table 7 sensors-22-09288-t007:** Fitting and determination coefficient for the JXCT probes when increasing the nitrate concentration in sand at 35% volumetric soil moisture.

JXCT Probe	a	b	R^2^
	Y = ax + b		
Probe 1	6.47	−283.35	0.93
Probe 2	7.01	−304.09	0.73

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
