# Peer review of "Evaluation of Nitrate Soil Probes for a More Sustainable Agriculture"

_sensors, 2022, doi:10.3390/s22239288_

Round 1

Reviewer 1 Report

This research was designed to examine the quality of three probes for estimating nitrate concentration at the soil matrix. After carefully reading through the manuscript, I found that the experiments were well organized, the conclusions were meaningful, and the topic falls within the scope of the journal. Nevertheless, it would be great if you authors could further polish the writing for improved readership. Below, I only provided a few minor comments.

Minor Comments:

Line 22: please be more specific about the meaning of “consistent results”.

Line 94: Period lost.

Section 2.1: Although it could be difficult to acquire the parameters of Rika and JXT, it would be better if you could add more information regarding these two sensors. This is because the current arguments in section 2.1 are not balanced since the description about Nutrisens is very complete.

Eq. 1: Either you can add a period after the equation or using lowercase first character for the word "where".

Table 2: The table look chaotic due to the equations. You may rearrange two equations and add more information.

Lines 373-374: There is a typo.

Reviewer 2 Report

The current article is entitled “Evaluation of nitrate soil probes for a more sustainable agriculture.” By Amelia Bellosta-Diest et al., studied to find out and analyze the performance of three commercial probes (Nutrisens, RIKA, and JXCT) under the same conditions.

Abstract:

•             This section seems unclear. The author should explain the results in detail and revise them in English native because there are a lot of problems with the language.

Introduction:

•             The significant novel point of the study over the precedent studies is not clear. More information should be added regarding whether choosing varied props within a suitable range of nitrate concentration in an aqueous solution and in the sand with 35% volumetric soil moisture.

•             9 references were found in the introduction (need more!), some of them are old and one is more than forty years ago! Replace them with a newer one!

There is no experimental design or statistical analysis for this study!

Results and discussion

The results and discussion section seem unclear. The author should explain the results in detail and revise them in English native because there are a lot of problems with the language.

In the results section, there is a striking lack of connectors between sentences and leading to confusion. In this section, the author should only concentrate on the underlying results rather than mentioning other’s studies.

Discussion is very shallow and needs in-depth discussion with the recent literature published. In discussion, there is a lack of a mechanistic approach.

There are many problems with the format and style, and please follow this journal format and style in the references! many of them missing a year of publication!

Reviewer 3 Report

This manuscript gives the study of 3 commericialized probes used for nitrate detection. The performance is discussed in terms of nitrate concentration, liquid/sand conditions. However, some parts need to be revised for this manuscript, which are bellows.

1, It is advised to mention the performance evaluation standards at the end of Introduction part.

2, For the data in table 1, it is advised to add the measurement precision.

3, For figure 8, the lengend colors are too similar, which is not friendly to the readers.

4, It is advised to add the range in terms of probe sensitivity discussion.
